# WHAT DOES YOUR BENCHMARK REALLY MEASURE? A FRAMEWORK FOR ROBUST INFERENCE OF AI CAPABILITIES

## ABSTRACT

Evaluations of generative models on benchmark data are now ubiquitous, and their outcomes critically shape public and scientific expectations of AI's capabilities. Yet skepticism about their reliability continues to grow. How can we know that a reported accuracy genuinely reflects a model's underlying performance? Although benchmark results are often presented as direct measurements, in practice they are inferences: treating a score as evidence of capability already presupposes a theory of what capability is and how it manifests in a testing environment.

We formalize this observation by proposing a principled framework that treats evaluation as inference: first, articulate a theory of capability, and then derive estimators that target this quantity. This perspective is well established in fields such as psychometrics but remains underdeveloped in AI evaluation, where implicit assumptions often go unexamined. As a proof of concept, we apply our framework to a concrete challenge that undermines reliability: model sensitivity to perturbations. We introduce several capability models and show how various sources of uncertainty (e.g., from finite samples and perturbations) arise within these models as nuisance terms of the latent capability itself. We then use standard tools to derive methods that infer capability while accounting for these sources of uncertainty. Our results illustrate how a capability-centered clarifies what evaluations measure and how to adjust for known sources of unreliability. More broadly, our framework yields evaluations that are transparent, grounded on cognitive theory, and better aligned with the scientific claims they aim to support.

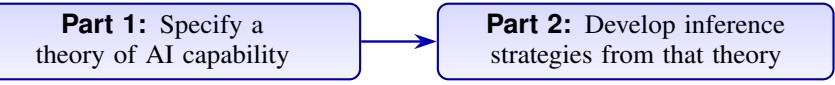

**Part 1:** Specify a theory of AI capability → **Part 2:** Develop inference strategies from that theory

## 1 INTRODUCTION

Evaluations (from hereon, "evals") of generative models have become ubiquitous as a way to probe each models' capabilities or harms. Companies developing large language models (LLMs) routinely assess their systems' intelligence using standardized knowledge tasks, while research papers proposing new methods often conduct comparative evaluations against state-of-the-art models. Leaderboards hosted on Vellum and Huggingface have also emerged as open-source platforms for directly comparing the capabilities of various LLMs. The rapidly growing interest in evals reflects our collective desire to understand how generative models behave, especially as they are now widely utilized, touted as highly capable, but inherently black box in nature.

In this work, we focus on evals that use standardized benchmark datasets. Despite the many criticisms of benchmarking (Alzahrani et al., 2024; Raji et al., 2021), benchmark data still comprise the vast majority of evals today. Benchmark results can carry significant influence in calibrating public and scientific expectations of AI capabilities[1], which motivates the need for robust and trustworthy methods for inferring and reporting these capabilities. Yet, there is growing consensus that generative AI evals using benchmarks are brittle and unreliable (Mitchell, 2023; Eriksson et al., 2025).

---

[1]Notably, companies that develop LLMs publish heavily cited reports on their models' performance on various benchmarks (Achiam et al., 2023; Team et al., 2023; Guo et al., 2025).

We believe two key reasons contribute to this lack of reliability. **First, most evals are not grounded in an explicit theory of capability.** Standard metrics such as accuracy are often treated as if they directly represent "what the model can do." But interpreting a benchmark score as evidence of capability already presupposes a theory of what the capability is and how it is expressed in a test. When this theory remains implicit, the connection between observable performance and underlying ability becomes unclear. This ambiguity is especially problematic given, for example, substantial evidence that small perturbations to phrasing or structure can significantly alter model outputs (Mizrahi et al., 2024; Sclar et al., 2023; Zhuo et al., 2024; Zheng et al., 2023; Errica et al., 2024). Without a theory specifying which behavioral features constitute capability, it is unclear how such sensitivity should be interpreted or corrected.

**Second, evals rarely quantify uncertainty in a way that corresponds to what capability is intended to capture.** Existing treatments, when they occur, focus primarily on finite-sample uncertainty (Chiang et al., 2024; Miller, 2024). But generative model evals face deeper uncertainties, such as perturbation sensitivity and contextual instability. The relevance of each source depends on one's theory of capability. When that theory is left implicit, uncertainty is modeled incompletely, and important contributors to unreliability are overlooked.

We argue that evals for generative models must return to their conceptual roots in statistical inference and begin by defining what capability means. We take inspiration from psychometrics and educational assessments, where statistical models have long been used to infer latent human abilities from observed performance. Following this tradition, we aim to articulate a theory of AI capability that identifies the multiple sources of uncertainty that threaten inference, and to derive estimators of capability whose form follows directly from this theory.

**Contributions.** *(1)* We present a conceptual argument. We begin by showing that every evaluation implicitly embeds a theory of capability – even when unstated – and that different theories imply fundamentally different interpretations of what "capability" means (Section 3). Then, we argue that these theories, being derived from *human* behavior, must be amended to model non-human patterns of AI behavior, such as brittle semantic generalization (Section 4).

*(2)* In response to the problem we identified, we develop a *principled framework* for evaluating generative models by treating benchmark evaluation as an inferential problem in Section 5. The core idea is to **begin with a formal theory of capability and derive the corresponding inference procedure**. This shift clarifies what evaluations should measure and which statistical assumptions are required for the resulting estimates to be meaningful. As a proof of concept, in Section 6.1, we apply our framework to a crucial problem confounding evaluations: *sensitivity to perturbations*. These examples illustrate how a theory-first perspective naturally exposes the assumptions behind evaluation and highlights where principled adjustments are needed.

*(3)* Building on this theoretical foundation, in Section 6.2, we present four inference methods corresponding to different capability constructs. These inference methods are derived from *standard statistical machinery*, and demonstrates the ease of deriving inference methods once a theory of capability has been established. In general, **there is no free lunch** in that all capability notions and inference methods face trade-offs on sample complexity, structural assumptions, etc.

## 1.1 RELATED WORK

**Perspectives on AI benchmarking.** A broad literature highlights conceptual and methodological gaps in how AI evaluations are designed and interpreted. Psychometrics provides a mature foundation for formalizing constructs such as *validity* – that evaluations properly measure a construct of capability – and *reliability* – that evaluations yield measures that are replicable and consistent (Lord, 1980; Raykov & Marcoulides, 2011). Several recent works argue that AI benchmarking would benefit from similar principles (Wang et al., 2023; Raji et al., 2021), while others critique how benchmarks are built, saturated, and deployed (Kiela et al., 2021; Bowman, 2023; Dehghani et al., 2021). Our contribution extends this line of work by making capability assumptions explicit, modeling the sources of uncertainty that undermine validity, and treating benchmark evaluation as a principled inferential task.

**Skills and artificial general intelligence.** There is increasing interest in quantifying artificial general intelligence, which could mean either high performance across diverse tasks (Hernández-Orallo et al., 2021) or the ability to accumulate and recombine skills (Chollet, 2019). Many probes on this

question often focus on large composite benchmarks such as MMLU (Hendrycks et al., 2021) and AGIEval Zhong et al. (2023), without consideration on how specific questions or tasks map into some latent notion of intelligence. Our framework clarifies how different capability models encode a particular skill structure, offering a principled link between benchmark questions/tasks and latent skill hierarchies relevant to discussions of artificial general intelligence.

**Robustness.** Across modalities, modern AI systems exhibit marked sensitivity to even *non-adversarial* perturbations. In vision, small semantic-preserving changes can shift predictions due to reliance on non-robust features (Ilyas et al., 2019); in language, LLMs vary unexpectedly under prompt rephrasings, formatting changes, or answer-order variations (Zheng et al., 2023; Sclar et al., 2023; Mizrahi et al., 2024; Zhuo et al., 2024). While related, we make broader argument that there is mismatch between how AI models behave in testing environments and the behavioral assumptions inherited from human-centered evaluation theory – many of which involves robustness issues.

## 2 THE MANY THEORIES OF CAPABILITY

In this section, we present several distinct theories of "capability" used across psychometrics, educational assessments, and adjacent fields. Each offers different structural assumptions and trade-offs.

**(a) Classical Test Theory (CTT).** Originating in the early 20th century, CTT models an observed test score $\phi$ as the sum of a true score $\theta$ and random error $\epsilon$ (Raykov & Marcoulides, 2011):

$$\phi = \theta + \epsilon. \tag{1}$$

Assumption 1 requires that $\epsilon$ has mean zero and is independent of $\theta$, reflecting the idea that humans make *random, ability-independent* mistakes[2]. Under parallel forms of a test, repeated scores converge to $\theta$ (Lord & Novick, 2008).

**Assumption 1.** *Under the CTT model in* (1), $\mathbb{E}[\epsilon] = 0$ *and* $\mathrm{Cov}(\theta, \epsilon) = 0$.

**(b) Item Response Theory (IRT).** Modern assessment adopts latent-trait models in which a $K$-*dimensional* ability vector $\theta \in \mathbb{R}^K$ governs success probabilities. A Rasch model specifies

$$f_i(\theta) \;=\; \Pr(\phi_i = 1 \mid \theta) \;=\; \sigma(a_i^\top \theta - b_i), \tag{2}$$

where $a_i \in \mathbb{R}^K$ are item loadings and $b_i$ is item difficulty. Observed responses are Bernoulli draws,

$$\phi_i = f_i(\theta) + \epsilon_i, \tag{3}$$

where $\epsilon_i$ represents logistic or probit noise. By modeling how items discriminate along dimensions in $\theta$, IRT provides sample-efficient estimates, and underlies much of today's adaptive standardized tests (College Board, 2025). IRT also satisfies Assumption 1; see Proposition 2.

**(c) Cognitive Diagnostic Models (CDM).** CDMs represent capability as a vector of (typically) discrete skill masteries (Leighton & Gierl, 2007). Let $\alpha \in \{0, 1\}^K$ encode whether a test-taker has mastered each of $K$ underlying skills, and let $Q \in \{0, 1\}^{m \times K}$ map items to the skills they require. Models such as DINA or DINO specify

$$f_{\mathrm{CDM}}(\alpha, Q_i) = \Pr(\phi_i = 1 \mid \alpha) = g_i + (1 - g_i)\mathbf{1}\{\alpha \text{ satisfies } Q_i\},$$

with slip and guess parameters $(s_i, g_i)$. CDMs thus interpret capability as *which* skills are possessed.

**(d) Bayesian Network Skill Models (BNSM).** In intelligent tutoring systems, capability is often modeled as a structured latent state evolving over a graph of prerequisite relations. Skills form nodes of a Bayesian network, and each skill variable $S_k$ has a posterior $\Pr(S_k = 1 \mid \text{data})$ updated by item responses (Culbertson, 2016). Items provide noisy evidence via conditional probability distributions (CPDs), and inference propagates beliefs across the graph. This yields a capability representation that is *structured*, accommodating both conceptual dependencies and temporal updates.

Figure 1 illustrates the implicit generative structure assumed by the various theories of capability. Note that this list is non-exhaustive. For example, there are nonparametric item response models such as Mokken scaling that drop parametric assumptions (Mokken, 1971; Sijtsma & Molenaar,

---

[2]CTT additionally assumes independence of errors across questions. While strong for humans, modern generative models often approximate this property, making it less restrictive in AI evaluation.

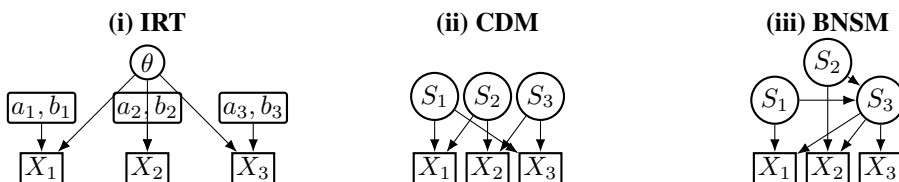

Figure 1: Structural hierarchy across models. **(i)** IRT: a single latent ability $\theta$ generates all item responses. **(ii)** CDM: multiple skill nodes $\{S_k\}$ connect to items according to a fixed $Q$-matrix. **(iii)** BNSM: skills form a directed concept hierarchy with interdependencies among $\{S_k\}$.

2012) and dynamic learning models such as Bayesian Knowledge Tracing and performance factor analysis that treat capability as evolving over time (Corbett & Anderson, 1994; Pavlik et al., 2009). In psychophysics and decision science, ability is operationalized via signal detection sensitivity or drift-diffusion parameters (Green & Swets, 1966; Ratcliff & McKoon, 2008). No single theory is strictly better; each embodies trade-offs between interpretability, statistical and behavioral assumptions, and the kinds of constructs they can represent, see e.g., Table 3.

## 3 EVALS IMPLICITLY ASSUME A THEORY OF CAPABILITY

### 3.1 MOST ACCURACY-BASED EVALS INSTANTIATE CTT

In practice, CTT reduces to averaging correctness across items, implicitly assuming that all questions are equally informative about capability. This mirrors nearly all contemporary AI evaluation: benchmarks report aggregate accuracy, pass rates, or score averages, with no item-level parameterization. Formally, letting

$$\phi_i = \theta_i + \epsilon_i, \tag{4}$$

CTT treats each observed response as an unbiased but noisy measurement of a per-item score $\theta_i$, yielding overall capability $\theta = \mathbb{E}_i[\theta_i]$. This assumption family underlies virtually all widely used static benchmarks across NLP, vision, and reasoning tasks: a model's "capability" is its expected accuracy under an item distribution.[3]

### 3.2 RECENT IRT-BASED METHODS IMPLICITLY ADOPT A DIFFERENT THEORY OF CAPABILITY

A growing line of work proposes IRT-inspired methodologies for AI evaluation, often to reduce sample complexity. These approaches replace the CTT viewpoint with a *latent trait* model in which a single (or low-dimensional) ability parameter $\theta$ generates response probabilities as in (3). Examples include an adaptive testing design (Zhuang et al., 2023), constructing smaller, more informative benchmarks (Maia Polo et al., 2024), and others (Burdick et al., 2019; Hernandez et al., 2021; Dong et al., 2020; 2021; Wang et al., 2025; Chen et al., 2025).

While powerful, these works instantiate a *different* theory of capability: ability $\theta$ is a latent parameter defined by a generative model of item responses, not the empirical accuracy of CTT. Under IRT, two models with the same accuracy can receive different ability estimates if their errors fall on items with different discrimination parameters. Without making the underlying theory explicit, such differences in what "capability" means can easily lead to misinterpretation and inconsistent comparisons.

## 4 WHY EXISTING CAPABILITY THEORIES DO NOT APPLY TO AI

All existing theories in Section 2 were developed to describe *human* test-taking behavior. Despite modeling differences, these traditions share a common foundation: latent ability (or skill mastery) generates systematic performance, while deviations from that structure are treated as *random, ability-independent*, and typically *independent across items*. In CTT this is explicit through Assumption 1; in IRT, CDM, and BNSM, it appears through the assumption that conditional on the latent trait, item responses are independent and error terms are unbiased. These assumptions reflect empirical regularities of human cognition, but they are systematically violated by current AI systems.

---

[3]This also includes evaluate-once leaderboards, security benchmarks, and multi-skill exams where per-task scores are averaged; all such summaries collapse to CTT under Assumption 1.

A large body of evidence shows that AI models often fail in ways that humans do not: they exhibit "Potemkin" or superficial understanding (Mancoridis et al., 2025), brittle semantic generalization (Mizrahi et al., 2024; Sclar et al., 2023; Zheng et al., 2023), and sensitivity to superficial rephrasings, distractors, and formatting (Zhuo et al., 2024; Errica et al., 2024; Du et al., 2022). Moreover, model behavior depends strongly on hyperparameters (e.g., temperature, top-$p$), system prompts, and surrounding conversational context. These behaviors contradict the foundational assumption shared across human-centric capability theories: that errors represent noise rather than systematic, context-driven shifts in performance.

To illustrate, consider a more realistic model of AI capability *within the CTT paradigm*:

$$\phi_i = \theta_i + s(x_i) + r(h) + g(c) + \cdots + \epsilon_i, \tag{5}$$

where $x_i$ captures input features of item $i$ (e.g., phrasing or structure), $h$ denotes hyperparameters (e.g., sampling temperature), and $c$ encodes contextual or environmental variables (e.g., system prompts). The functions $s$, $r$, and $g$ represent *systematic*, non-random performance shifts. When $x_i$, $h$, or $c$ vary across evaluation settings, these structured biases confound estimation of the underlying item-level capability $\theta_i$. The ellipsis indicates additional confounding factors.

## 5 FRAMEWORK FOR ROBUST INFERENCE OF CAPABILITIES

The previous sections showed that AI evaluations implicitly rely on *incompatible* theories of capability: accuracy-based benchmarks instantiate CTT (Section 3.1) while recent adaptive methods adopt IRT (Section 3.2). This conceptual inconsistency contributes to the "wild west" nature of evals and makes reliability and comparability difficult. We propose a simple framework that unifies the theoretical grounding of evaluations:

**Part 1:** Specify and defend a theory of AI capability → **Part 2:** Develop inference strategies from that theory

**(1) Evals must explicitly state their theory of capability.** A benchmark's underlying theory determines *what* its scores mean, *which comparisons are valid*, and *how results should generalize across tasks or settings*. Without stating the underlying theory, comparisons across benchmarks—or even across modeling choices within the same benchmark—become conceptually undefined. In Section 6.1, we define various theories of capability as a proof-of-concept.

**(2) Inference methods come from the theory.** Given a theory of capability, inference methods often come naturally using standard statistical tools. We demonstrate this in Section 6.2.

## 6 PROOF-OF-CONCEPT: TACKLING SENSITIVITY TO PERTURBATIONS

### 6.1 PART 1: THEORIES OF AI CAPABILITY

| Model family | Functional form |
|---|---|
| **CTT** | $\phi_i = \theta_i + s(x_i) + \epsilon_i$ |
| **IRT (1PL)** | $\phi_i = \sigma(\theta - b_i) + s(x_i) + \epsilon_i$ |
| **Cognitive-Diagnostic Models** | $\phi_i = f_{\text{CDM}}(\alpha, Q_i) + s(x_i) + \epsilon_i$ |
| **Bayesian Network Skill Models** | $\phi_i = \Pr(\phi_i = 1 \mid S, \text{graph}) + s(x_i) + \epsilon_i$ |

Table 1: Functional forms of four capability theories, augmented with a perturbation term $s(x_i)$ capturing systematic shifts due to variations in input phrasing or structure.

**Assumption 2** (Mean-zero perturbations). *Let $\mathcal{P}_i$ denote the distribution of natural perturbations of question $i$. Then*

$$\mathbb{E}_{x_i \sim \mathcal{P}_i}[s(x_i)] = 0.$$

Table 1 presents several examples of theories of capability. Each theory requires Assumption 2, which states that capability is recovered only *in expectation* over the distribution of "natural perturbations".

Without this assumption, latent traits become unidentifiable: the perturbation function $s(\cdot)$ can absorb item properties, trait parameters, or both. **Explicitly stating the theory of capability makes these crucial assumptions transparent.**

### 6.1.1 BENCHMARK CURATION VIOLATES AN INDEPENDENCE ASSUMPTION

We focus on the CTT model for clarity, though a similar argument holds trivially for the other models in Table 1. Let $\mathcal{D} = \{x_i\}_{i=1}^n$ denote a benchmark. Conceptually, generating an item $x_i$ involves two stages: *Stage 1 (Question sampling):* Draw a question or concept $i$ from a latent distribution over the task space $\mathbb{P}$. *Stage 2 (Phrasing sampling):* Draw a natural phrasing $x_i$ for that question from the high-dimensional, unknown phrasing distribution $\mathcal{P}_i$.

Benchmark curators effectively control Stage 1 through their choice of questions, which can be viewed as independently sampled from an implicitly defined $\mathbb{P}$. However, curators do *not* observe or sample from the true $\mathcal{P}_i$. In practice, each question receives only a single, hand-designed phrasing $x_i$, and these phrasings are produced by the same individuals or pipeline, introducing stylistic and structural dependencies across items. Thus, benchmarks almost always produce *dependently sampled* draws from $\mathcal{P}_i$, violating Assumption 2. This makes it impossible to identify $\theta_i$ under Table 1 (top row), even though identification is trivial under the classical CTT model (4). Empirically, this manifests as different or conflicting inferences on accuracy, see Appendix C.

### 6.1.2 PERTURBATIONS FOR PSEUDO-INDEPENDENCE

A natural response to the dependence problem is to approximate $\mathcal{P}_i$ by generating multiple phrasings for each question. Prior work already follows this intuition: perturbing instructions (Mizrahi et al., 2024), question wording (Sclar et al., 2023), or answer ordering (Zheng et al., 2023) all implicitly aim to sample from a richer portion of $\mathcal{P}_i$. Our framework clarifies that these methods are attempts to recover identifiability by increasing coverage of the phrasing space.

Let $\tilde{\mathcal{D}} = \{\{x_{ij}\}_{j=1}^{m_i}\}_{i=1}^n$ be a perturbed benchmark, where each $x_{ij}$ is produced by a perturbation mechanism intended to approximate draws from $\mathcal{P}_i$. The CTT model then becomes $\phi_{ij} = \theta_i + s(x_{ij}) + \epsilon_{ij}$. Although perturbation generators can never match the true (and fundamentally unknowable) $\mathcal{P}_i$, perturbations may improve identifiability, as in the following result.

**Proposition 1.** *Let a benchmark contain phrasings drawn from an unknown distribution $\mathcal{P}_i^{(0)}$. Let $\delta_i^{(0)} := \mathbb{E}_{\mathcal{P}_i^{(0)}}[s(x)]$ denote the induced bias in the recovered latent trait. Suppose a perturbation mechanism generates $m_i$ variants $\{x_{ij}\}_{j=1}^{m_i}$ with distribution $\tilde{\mathcal{P}}_i$ and bias $\delta_i := \mathbb{E}_{\tilde{\mathcal{P}}_i}[s(x)]$.*

*Define the plug-in estimator $\hat{\theta}_i := \frac{1}{m_i} \sum_{j=1}^{m_i} \phi_{ij}$. Then:*

    *(i) As $m_i \to \infty$,   $\hat{\theta}_i \xrightarrow{a.s.} \theta_i + \delta_i$.*

    *(ii) $|\delta_i| < |\delta_i^{(0)}|$   and   $\mathbb{E}\left[(\hat{\theta}_i - \theta_i)^2\right] < \mathbb{E}\left[(\phi_i^{(0)} - \theta_i)^2\right]$ if $dist(\tilde{\mathcal{P}}_i, \mathcal{P}_i) < dist(\mathcal{P}_i^{(0)}, \mathcal{P}_i)$.*

**The perturbation mechanism remains a modeling decision.** Proposition 1 states that perturbations cannot remove bias unless $\tilde{\mathcal{P}}_i = \mathcal{P}_i$ almost surely, but they strictly reduce both bias and variance whenever they expand coverage of the phrasing space. Proofs can be found in Appendix B.

## 6.2 PART 2: INFERENCE STRATEGIES

Once a theory of capability is fixed, **inference strategies often follow from standard statistical results**. In this section, we demonstrate how different theories of capability naturally induce different estimands and inference pipelines, following the four theories in Section 6.1.

### 6.2.1 CTT: ESTIMATING ACCURACY VIA CLUSTERED BOOTSTRAPPING

Under the perturbed CTT model, the estimand is $\theta = \frac{1}{n} \sum_{i=1}^n \theta_i$. We estimate item-level accuracy by averaging over perturbations, $\hat{\theta}_i = \frac{1}{m_i} \sum_j \phi_{ij}$, and estimate $\theta$ by the sample mean $\hat{\theta} = \frac{1}{n} \sum_i \hat{\theta}_i$. Standard CLT justifies asymptotic normality, but estimating population variance is difficult; we

**(a) CTT: Accuracy Estimation (CBA)**
**Require:** Item clusters $\{x_{ij}\}$, $m_i$ perturbations
1: For each item $i$: $\hat{\theta}_i \leftarrow \frac{1}{m_i} \sum_j \phi_{ij}$
2: Estimate accuracy: $\hat{\theta} \leftarrow \frac{1}{n} \sum_i \hat{\theta}_i$
3: Cluster bootstrap items $\{1, \ldots, n\}$ to obtain CI
4: **return** $(\hat{\theta}, \text{CI})$

**(b) IRT: Adaptive Ability Estimation (LAAT)**
**Require:** Item params $(a_i, b_i)$, prior $\theta_0$
1: Initialize $\theta \leftarrow \theta_0$, info $\mathcal{I} \leftarrow 1/\sigma_0^2$
2: **while** not converged **do**
3:     Select $i^* = \arg\max_i I_i(\theta)$ (Fisher info)
4:     Query $m$ perturbations; $\phi_{i*} \leftarrow \frac{1}{m} \sum_j \phi_{i*j}$
5:     Update via Newton step: $\theta \leftarrow \theta + S/\mathcal{I}$
6: **end while**
7: **return** $(\theta, 1/\sqrt{\mathcal{I}})$

**(c) CDM: MAP Skill Estimation**
**Require:** $Q$-matrix, link $f_{\text{CDM}}$, perturbations $\{x_{ij}\}$
1: Aggregate items: $\bar{\phi}_i = \frac{1}{m_i} \sum_j \phi_{ij}$
2: Form log-posterior $\log p(\theta) + \sum_i \log p(\bar{\phi}_i \mid \theta, Q_i)$
3: $\hat{\theta} \leftarrow \arg\max_\theta$ of the log-posterior (e.g. via Newton/L-BFGS)
4: Bootstrap items to obtain skill-level uncertainty
5: **return** $\hat{\theta}$ and posterior skill summaries

**(d) Bayesian Network Skills: Posterior Inference**
**Require:** BN structure, Gaussian skill prior, logistic item CPDs, $\{\phi_{ij}\}$
1: Aggregate evidence: $\bar{\phi}_i = \frac{1}{m_i} \sum_j \phi_{ij}$
2: Run BN inference (continuous–discrete belief propagation)
3: Compute posterior $p(S \mid \{\bar{\phi}_i\})$
4: Optionally bootstrap items for uncertainty
5: **return** Posterior mastery $\{\sigma(S_k)\}$

Table 2: Abridged inference algorithms for the four theories of capability introduced in Section 2. These methods follow directly from standard statistical machinery. Detailed algorithms can be found in Appendix D.1.

therefore use a *clustered bootstrap* (items as clusters, perturbations within clusters) (Ren et al., 2010; Field & Welsh, 2007). The procedure is shown in Table 2(a) and in detail in Algorithm 1.

### 6.2.2 IRT: ESTIMATING LATENT ABILITY VIA ADAPTIVE TESTING

Under the perturbed IRT model, the estimand is a latent ability $\theta_k$ for each model $k$. Once item parameters $(a_i, b_i)$ are calibrated, estimating reduces to classical IRT inference using Fisher scoring and Newton–Raphson updates (Raykov & Marcoulides, 2011):

$$\theta \leftarrow \theta + \frac{S(\theta)}{\mathcal{I}(\theta)},$$

where $S$ and $\mathcal{I}$ are the score function and the observed Fisher information for the IRT likelihood. To reduce sample complexity, we apply *adaptive item selection*: at each step, choose the item with maximal Fisher information at the current estimate. The adaptive test is summarized in Table 2(b) and in detail in Algorithm 2.

### 6.2.3 CDM: ESTIMATING SKILL VECTORS VIA MAP

In CDMs, capability is typically a binary skill vector $\alpha \in \{0, 1\}^K$. Here, we relax the binary assumption to $\alpha \in \mathbb{R}^K$ to get better-calibrated results. Each question loads on a subset of skills via the $Q$-matrix and follows a logistic response model $f_{\text{CDM}}(\theta, Q_i)$. Perturbations are aggregated as $\bar{\phi}_i = \frac{1}{m_i} \sum_j \phi_{ij}$. Inference reduces to penalized maximum a posteriori (MAP) estimation:

$$\hat{\theta} = \arg\max_\theta \left\{ \sum_i \log p(\bar{\phi}_i \mid \theta, Q_i) + \log p(\theta) \right\},$$

typically using a Gaussian prior over $\theta$. Item-level bootstrapping yields uncertainty over each skill dimension. The procedure is summarized in Table 2(c), and in Algorithm 3.

### 6.2.4 BNSM: ESTIMATING SKILL VECTORS VIA POSTERIOR INFERENCE

In BNSMs, the vector $S \in \{0, 1\}^K$ captures mastery of $K$ skills. We generalize to continuous values by modeling skills as correlated Gaussian latent nodes, with items as Bernoulli children governed by logistic CPDs. Aggregated perturbations $\bar{\phi}_i$ provide conditionally independent evidence for each item. Given the BN structure and parameters, posterior inference proceeds via standard continuous-discrete belief propagation $p(S \mid \{\bar{\phi}_i\})$, from which we extract posterior skill mastery (e.g. $\sigma(S_k)$). Bootstrapping items again provides uncertainty. See Table 2(d) for pseudo-code, and

| | Classical Test Theory | 1-Dimensional Item Response Theory | Cognitive Diagnostic Models | Bayesian Network Skills |
|---|---|---|---|---|
| **Interpretation of ability** | Accuracy over dataset | Latent ability (relative to prior) | Mastery of $K$ skills | Posterior prob. of mastery of $K$ skills |
| **Structural asn.** | None | See Figure 1(i) | See Figure 1(ii) | See Figure 1(iii) |
| **Additional data needed** | None | Item parameters $(a_i, b_i)$ | Q-matrix $Q$; CDM specification | BN structure + CPTs |
| **Inference method** | Clustered bootstrap over items | Adaptive test (Fisher info + Newton–Raphson) | MAP over $2^K$ skill profiles (binomial CDM likelihood) | Belief propagation / BN posterior computation |
| **Budget behavior** | Fixed benchmark | Adaptive selection due to pre-calibration | Can be adaptive in theory (requires expensive calibration) | |

Table 3: Comparison across four theories of capability/inference methods. Each comes with trade-offs on interpretability, additional data required, structural or functional assumptions, etc.

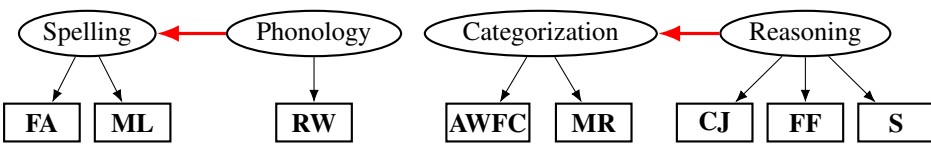

Figure 2: Skill structure for CDM and BNSM. Black arrows show the skill–task mapping. Red arrows show additional conceptual structure used in BNSM.

in Algorithm 4. Like IRT, CDM & BNSM can in theory support adaptive testing, but only after a calibration phase that jointly estimates item parameters and the latent skill dependencies.

### 6.3 EMPIRICAL STUDY

**Setup.** We evaluate seven open-source instruction-tuned LLMs (Llama-3.2, Qwen-2.5, and Gemma families) on two benchmarks, Big-Bench Hard (BBH) (Suzgun et al., 2023) and LMEntry (Efrat et al., 2023), both of which have perturbed versions from (Mizrahi et al., 2024). Each dataset contains sub-tasks testing different concepts, and we use four from each category. For LMEntry, we use `any word from category` (**AWFC**), `first alphabetically` (**FA**), `more letters` (**ML**), and `rhyming word` (**RW**). For BBH, we use `causal judgment` (**CJ**), `movie recommendation` (**MR**), `formal fallacies` (**FF**), and `snarks` (**S**). See Appendix C.1 for details on the tasks, perturbations, and evaluation procedure.

**Skill structure.** For CDM and BNSM, we specify a skill mapping as in Figure 2. This is a simple choice of many, but serves as a demonstration of the structural flexibility of CDM and BDSM.

Figure 3 show estimates of capability from the four methods introduced in Table 2 over seven LLMs. We show the full suite of results in Appendix E.2. Generally, the ordering in model rankings is consistent between the methods. However, LAAT yields more separation between models when bootstrapping cannot. For example, `Qwen-3.5B` on **S** benchmark has the highest inferred performance using both methods, but LAAT infers much higher ability for that model compared to the rest because it happens to perform well on harder questions. Meanwhile, CDM and BNSM are able to aggregate higher-order skills based on the structural assumptions we made about models' relevant skills. BNSM and CDM results do not vary significantly because of the similarity in the skill mapping (Figure 2).

## 7 DISCUSSION

### 7.1 TAKEAWAYS AND CONCRETE RECOMMENDATIONS

**Explicitly state the theory of capability used in any evaluation.** Different benchmarks implicitly assume different theories (CTT, IRT, CDM, Bayesian skill models). Making these assumptions

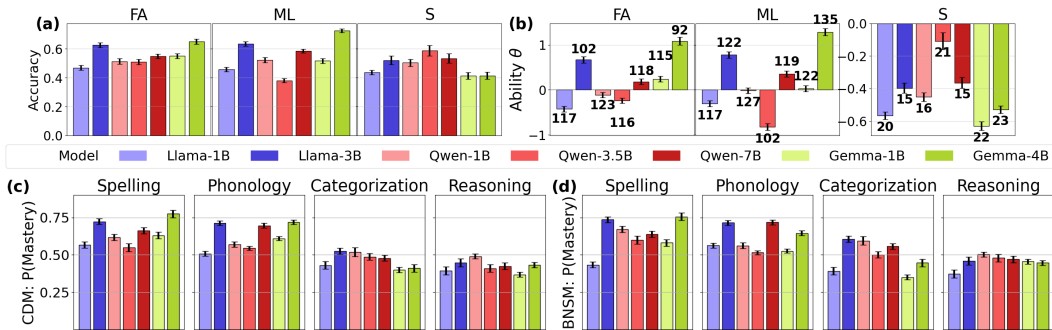

Figure 3: (a) Estimates of accuracy under CTT (Alg. 1, (b) Estimates of ability under IRT (Alg. 2), (c) and (d) Estimates of mastery of skills under CDM (Alg. 3) and BNSM (Alg. 4), respectively, as specified by the structure in Fig. 2. These theories and methods explicitly account for model sensitivity to perturbations, see Table 1. We test seven open-source LLMs on eight benchmark tasks across LMEntry and BBH (full results shown in Appendix Figures 8 and 9). Numbers in bold indicate number of questions asked in the adaptive test. For the other methods, the number of questions asked is roughly 500. Each question is associated with 20 random perturbations.

explicit clarifies what the reported quantity *means*, what assumptions are required (e.g., mean-zero phrasing effects), and what sources of variation are treated as noise versus structure. Stating the theory calibrates expectations and prevents misinterpretation of the resulting scores.

**Do not compare quantities across different theories of capability.** Accuracy under CTT, latent ability under IRT, discrete skill profiles under CDMs, and posterior mastery under Bayesian models represent *different constructs*. They are not interchangeable. Claims of superiority or comparisons across systems are only meaningful when they rely on the *same* underlying capability theory.

**All theories and inference strategies face trade-offs.** Table 3 summarizes how each theory differs from each other in terms of data needed and the assumptions needed. In general, *there is no free lunch*. For example, CTT offers a simple, widely-understood interpretation (accuracy) but requires evaluating models on every item to control variance. IRT provides a more conceptual quantity (ability) and can reduce sample complexity via adaptive designs, but depends on correctly calibrated item parameters (difficulty, discrimination). CDM and Bayesian skill models offer richer structural interpretations at the cost of stronger modeling assumptions.

## 7.2 LIMITATIONS AND FUTURE WORK

**Other issues confounding evaluations.** While our proof-of-concept centers on sensitivity to phrasing perturbations, many additional confounders remain (Section 5), such as hyperparameter choices, system prompts, and evaluation context.

**Construct validity remains a crucial issue.** Our work mainly focuses on benchmark *reliability* rather than ensuring that benchmarks truly assess the construct of interest to begin with. Perhaps benchmarks may be fundamentally limited in this regard (Raji et al., 2021), but psychometrics may provide more insights to improve the construct validity of benchmarks (Wang et al., 2023).

**Assumptions about AI behavior remain speculative.** Choosing any theory of capability—CTT, IRT, CDM, or BNSM—imposes assumptions about how AI systems behave (e.g., stability across perturbations, monotonicity, item invariance). These assumptions were historically motivated by human cognition, not generative models, and may be misspecified or incomplete. As our understanding of model behavior improves, so too must the underlying theories and inference procedures.

For instance, defining constructs such as item difficulty or latent ability raises deep conceptual questions. What does it mean for a problem to be intrinsically difficult for an AI system? Should difficulty and ability be benchmark-relative or universal? Recent work suggests the possibility of a uni-dimensional intelligence factor for LLMs (Ilić & Gignac, 2024), prompting new questions about how such a quantity should be tested and robustly measured. We view our framework as contributing to a broader effort to build the science of benchmarks (Hardt, 2025) and the measurement theory of artificial general intelligence (Mitchell, 2024).

## REPRODUCIBILITY STATEMENT

Proofs and additional theorems can be found in the Appendix. The code and datasets used to produce the results presented in the paper can be found in the supplementary material, or at https://anonymous.4open.science/r/ai_stat_test-2100/.

## USE OF LLMs

LLMs were used to edit and polish some parts of the writing.

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
