# OpenReview forum: "What Does Your Benchmark Really Measure? A Framework for Robust Inference of AI Capabilities"
_ICLR.cc/2026/Conference — ICLR 2026 Conference Desk Rejected Submission_

### Official Review · Reviewer_iZmD · 2025-10-24

**Soundness:** 4
**Presentation:** 3
**Contribution:** 2
**Rating:** 4
**Confidence:** 4

**Summary:**

The paper attempts to take an inference-driven approach to AI evaluation --- defining AI capabilities explicitly with a theory of their operation and then design tests that allow inference of capability levels. This is explored across  systematic bias experiments, and with adaptive testing, revealing that systems can be evaluated with reduced sample complexity.

**Strengths:**

The paper is generally rigorous and well-written.
I particularly like the argument about how benchmark creation violates an independence assumption. I think this is a valuable point to emphasise more in the paper and the implications it has on evaluation---namely the difficulties in accurately measuring abilities.

The work in section 4.3 on systematic bias is really strong, and I think makes a good argument for fixing the independence violation in terms of estimating accurate ability score on benchmarks. It would have been even stronger if a wider range of popular benchmarks was evaluated rather than the small collection here.

**Weaknesses:**

The paper presents the psychometric-inspired approach as if it were a novel contribution within the context of AI evaluation. However this isn't the case -- multiple papers have built on IRT before and the only one cited by the authors is more of a position paper about what's feasible with IRT. A few examples include:
- https://www.sciencedirect.com/science/article/pii/S0004370219300220
- https://www.ijimai.org/journal/bibcite/reference/2901
- https://openreview.net/pdf?id=ZyVQqK7mcP
- https://openreview.net/pdf?id=2QWP4qWVym
- https://arxiv.org/pdf/2503.06378?
- https://doi.org/10.1016/j.knosys.2021.108076
and more. These should be better cited as part of the existing literature to utilise psychometric approaches to AI evaluation.

The proposed response to fixing the dependence of sampling of the phrasing space uses phrasing perturbations. While I think this is broadly the right approach -- generate more test items that query the ability -- I don't think phrasing perturbations alone fix the problem. Often the constructs we want to evaluate require attention from many different angles. Approaches like Contrast sets https://arxiv.org/pdf/2004.02709 combined with the Factorial Survey method, provide better coverage of the construct than simply rephrasing the question. If by phrasing perturbations, things like contrast sets and Factorial Surveys are intended, the paper should more clearly reflect this and cite previous work more appropriately.

The work in 5.3 could be expanded in the main body of the text. It seems that this is the crux of the paper --- where the ideas are all supposed to come together to reveal the efficacy of the approach. However, the results are rushed through and largely relegated to the appendix, leaving the casual reader to likely miss the solid empirical work that has gone on here.

**Questions:**

1. What are the trade-offs between the CTT and IRT model in the empirical contexts? Is one more suited to AI evaluation over another?
2. Most of the novelty of this approach seems to be based on the bias term that is added (s(x_i)) to the capability model and handling that. Is there more novelty that I have missed or is that it? The algorithms developed in section 5 are, I believe, from the literature (or heavily based on existing work). Is this correct?
3. Why do you omit large sweathes of the literature that is already applying psychometric techniques to AI evaluation in the related work section?
4. Some models seem to be more susceptible to being systematically biassed against in the naive measurement vs the corrected measurement (Figure 2): What causes this to occur?

---

> ### Author Response · Authors · 2025-11-20
>
> Thank you for the positive comments. We appreciate the reviewer’s engagement and believe that several of the concerns stem from a misunderstanding of what the paper aims to contribute. Our goal is not to introduce novel algorithms but to formalize a framework for AI evaluation as statistical inference grounded in a theory of capability. The methods we present are intentionally simple, because they serve only as proofs-of-concept to illustrate how the framework operates.
>
> In the revision, we have **restructured the text to foreground the conceptual contribution** and clarify how existing tools naturally fall out once a capability model is made explicit. Below we address the specific concerns.
>
> **Weaknesses**
>
> P1. To clarify, our novelty is in framing benchmarks as an inference task that is grounded on a theory of capability. **Psychometry offers only some theories of capability (i.e., the CTT and IRT model), but there are others as well**. For example, educational testing has alternative models such as Cognitive Diagnostic Models (CDM) and Bayesian Network Skill Models (BNSM). In the revised draft (Section 2), we make clear that we pull from other relevant fields, those that propose some theory of (human) capability in the context of test-taking.
>
> If the reviewer is referring to the IRT-based model specifically, IRT is one of many ideas that originated from psychometrics, and our framework is much broader than that. We never suggest that the IRT-based method we propose is novel; in fact, in the previous draft we made this explicit in Section 5.2, citing some works that already do a similar approach. We appreciate the additional references, and we have cited them in Section 3.2 of the new draft. Our argument is precisely that these **IRT-based methods fall within our broader framework**, and that these works should be explicit about the theory of capability they use because it has a different interpretation to, say, CTT/accuracy.
>
> P2. The contrast set framing is interesting, but slightly tangential! Our perspective is that evaluating ability is about testing overall capabilities (over the population, rather than over a dataset). As such, we care more about the distribution or space of questions, rather than constructing questions that flip the model’s decision.
>
> P3. We would like to clarify that this is not our main contribution. Kindly refer to the first paragraph of our response.
>
> **Questions**
>
> Q1. Kindly refer to Table 1 and the Discussion for the trade-offs. We merely present these models as choices: some of many. In general, there is no free lunch in more robust evaluations. One needs a combination of (i) more data (e.g., perturbations or calibrated item set); or (ii) assume some more structure about the theory of capability.
>
> Q2. We would like to clarify that that is far from our main contribution; it is merely just an example to operationalize our framework! Kindly refer to the first paragraph of our response.
>
> Q3. Kindly refer to our response to P1.
>
> Q4. Great question! We also tried to answer this prior to submission, but found that the errors are largely idiosyncratic. This could be an interesting direction for future work, though we suspect that there is no discernible pattern to these biases in general.

---

> > ### Comment · Reviewer_iZmD · 2025-11-26
> >
> > Thank you for your response. To clarify my first weakness -- I mean that the idea of a unifying framework for capabilities isn't novel. I appreciate that you've taken the time to expand the options to include CDM and BNSM. However, I don't think your framework is ultimately unifying these concepts: The takeaway of the paper seems to be that these are distinct and shouldn't be "mixed and matched". I agree with you here -- there are different theories of capability and different measurement models, and these should be appropriately used and compared. I'm struggling to see why this constitutes a "novel framework" rather than simply saying "don't misuse measurement models".
> > Prior work arguably does a better job of unifying these concepts into a more comprehensive framework (von Davier, M. ,2008). How does your work relate to these General Diagnostic Models?
> >
> >
> >
> > von Davier, M. (2008). A general diagnostic model applied to language testing data. British Journal of Mathematical and Statistical Psychology, 61(2), 287–307. https://doi.org/10.1348/000711007X193957

---

> > > ### Author Response · Authors · 2025-11-26
> > >
> > > Thank you for taking the time to respond to our review!
> > >
> > > For one, von Davier (2008)'s main contribution is to unify various theories of capability (such as the IRT model) into a general class of diagnostic models. In the draft, we are not claiming to "unify" the various theories of capability, but merely presenting them as choices. Indeed, there could be some connections between the theories we laid out (as von Davier, 2008 points out), but that is not the point of our framework. We are simply **formalizing AI benchmarks as statistical inference tasks that are grounded on some theory of capability**, which is novel.
> > >
> > > Second, and perhaps more importantly, von Davier (2008) is mainly about inferring parameters of various diagnostic models in humans, **not in generative models**. A key part of our argument is that **generative models require different capability models**, see Section 4.
> > >
> > > We hope our response clarifies the novelty and usefulness of our framework. Please let us know if there are any misunderstandings or questions -- we would be happy to clarify further.

---

### Official Review · Reviewer_NPsR · 2025-10-29

**Soundness:** 2
**Presentation:** 3
**Contribution:** 2
**Rating:** 2
**Confidence:** 5

**Summary:**

This paper investigates current AI evaluations, questioning their reliability when treated as simple measurements. It proposes a new framework, inspired by psychometrics, that reframes evaluation as a formal statistical inference problem: one must first define a theory of AI "capability" and then derive methods to estimate it. As a proof of concept, the authors tackle sensitivity to input perturbations, empirically demonstrating that this factor introduces systematic bias into existing benchmarks, affecting even SOTA models. The paper introduces two robust inference methods: Clustered Bootstrapping to estimate true accuracy and a Latent Ability Adaptive Test based on Item Response Theory to efficiently estimate latent ability.

**Strengths:**

1. The paper investigates the reliability and trustworthiness of LLM benchmark evaluations, which is crucial for ensuring the scientific and healthy advancement of the AI field.

2. The paper reframes AI evaluation from a simple measurement to a formal statistical inference problem. This shift, inspired by psychometrics, provides a principled foundation to systematically address known issues like reliability and perturbation sensitivity.

3. The paper is clearly written and well-organized, the figures and tables are also clear and effectively support the paper's claims.

**Weaknesses:**

1. The paper's methodological contribution is incremental. The proposed methods (CBA and LAAT) are largely direct applications of standard techniques—Clustered Bootstrapping and Item Response Theory—from statistics and psychometrics. Given that prior work had already identified both the perturbation sensitivity problem and the potential use of IRT, the paper's primary contribution is its conceptual reframing, not a significant algorithmic advancement.

2. The LAAT algorithm's effectiveness is contingent on pre-inferred IRT item parameters $b_i, a_i$. The authors concede that reliably estimating these parameters requires a calibration step using a large sample of models, which implies a massive, upfront data collection effort. This requirement creates a significant cold-start problem, casting doubt on LAAT's practical utility for new or rapidly evolving benchmark tasks.

3. The paper's empirical support is insufficient to justify its broad claims. (1) The significant claim about SOTA model bias rests on only two models and two tasks, which is insufficient evidence for such a general claim. (2) The proposed methods (CBA and LAAT) are validated on only three tasks, which is not enough to demonstrate their general utility, especially given LAAT's high data prerequisites.

**Questions:**

Q1: The paper's core idea is to eliminate "systematic bias" by averaging over a large set of perturbations. This assumes the perturbation dataset itself is an unbiased sample. How do the authors guarantee that their perturbation-generation method does not simply introduce its own new systematic bias?

Q2: To provide accurate parameter estimates for the IRT model in LAAT, approximately what scale of interaction data is required? Furthermore, given this data prerequisite, can the LAAT method remain effective for the rapidly evolving new tasks that most require robust evaluation?

Q3: What are the basic assumptions of using the IRT model? Why did the authors not use Multidimensional IRT or other cognitive diagnostic models as the base model?

---

> ### Author Response · Authors · 2025-11-20
> **Misunderstanding of conceptual argument; significantly restructured paper to emphasize main contributions**
>
> Thank you for the thoughtful and detailed comments. We appreciate the reviewer’s engagement and believe that several of the concerns stem from a misunderstanding of what the paper aims to contribute. Our goal is not to introduce novel algorithms but to formalize a framework for AI evaluation as statistical inference grounded in a theory of capability. The methods we present are intentionally simple, because they serve only as proofs-of-concept to illustrate how the framework operates.
>
> In the revision, we have **restructured the text to foreground the conceptual contribution** and clarify how existing tools naturally fall out once a capability model is made explicit. Below we address the specific concerns.
>
> **Weaknesses**
>
> P1. We agree that the methodological components are incremental, and this is intentional. **The goal of the paper is not to introduce new algorithms but to demonstrate that evaluation procedures should follow from an explicit theory of capability**. Once the structure is specified, standard tools (bootstrapping, IRT) arise naturally. We clarified this explicitly in the revision.
>
> P2. Relatedly, LAAT is not presented as the most practical method. Its data requirements illustrate a central theme of our framework: more robust evaluation necessarily requires either (i) more data (e.g., calibrated item parameters or perturbations) or (ii) stronger modeling assumptions. There is no free lunch. Table 3 and the discussion (lines 458-464) now emphasize these trade-offs and situate LAAT (IRT) as one possible instantiation among many.
>
> In the discussion section (lines 475-485), we also point out that **any theory of capability is a choice**, and that there is no “correct” theory; that decision depends on the evaluator as long as they make them explicit in the benchmark. Future work should investigate AI’s cognitive behaviors to better inform the choice of capability theory in evaluations.
>
> P3. We have calibrated the scope of our empirical claims. The SOTA experiments are moved to the Appendix, where they are framed as probes that illustrate inconsistent inferences when perturbation sensitivity is ignored. Additional results on other benchmarks (Appendix F.2) are included, though we emphasize that these experiments are demonstrations of the conceptual issue, not a comprehensive empirical study.
>
> **Questions**
>
> Q1. Perturbation generation cannot be guaranteed to be unbiased; this is a fundamental limitation of any evaluation framework involving natural language variation. **Our contribution is to make this assumption explicit rather than implicit.** In our framework, perturbations define a phrasing distribution, and inference is only as reliable as the evaluator’s choice of that distribution. Section 6.1.2 now clarifies that any perturbation method induces its own phrasing space and that verifying its alignment with a “true” distribution is impossible. This is precisely why evaluations should be framed as inference, with assumptions stated transparently.
>
> Q2. IRT parameter estimation enjoys standard root-n behavior, but the more important requirement is population representativeness. Calibrating only on weaker or older models can distort item difficulties. LAAT is therefore most appropriate when a representative set of models is available—a point that reinforces the core message of our framework: different capability theories imply different data demands, and robustness must be understood through these trade-offs. We clarify this in the revision.
>
> Q3. Unidimensional IRT assumes local independence, monotonicity, and a single latent dimension. We used it because it is the simplest setting for demonstrating how latent-variable models structure evaluation and yield principled inference procedures. In the revision, Sections 2 and 6 explicitly discuss multidimensional IRT and cognitive diagnostic models as viable alternatives, and we emphasize that the models we use are illustrative examples, not prescriptive ones. The broader point is that inference methods follow naturally once the capability theory is chosen.
>
> Overall, **the revised draft clarifies that our contribution is the conceptual framework**: evaluations should articulate their capability assumptions, adapt these theories to AI behavior, and derive inference procedures from them. The specific methods in this paper are deliberately simple instantiations demonstrating how the framework operates.

---

### Official Review · Reviewer_rz2e · 2025-11-02

**Soundness:** 2
**Presentation:** 2
**Contribution:** 2
**Rating:** 4
**Confidence:** 3

**Summary:**

This paper reframes LLM benchmarking as inference rather than mere measurement: begin with a theory of capability, then derive estimators that adhere to that theory. It demonstrates that sensitivity to natural prompt perturbations systematically biases conventional accuracy-based scores and that even strong models (e.g., GPT-4.1) exhibit nontrivial bias; it further quantifies per-item variability via the mean absolute distance (M), with real-world deviations of 10–50 percentage points being common and original-versus-perturbed estimates differing by up to ~15 points. Building on this perspective, the authors propose two practical estimators: (a) clustered bootstrap over items to report accuracy with valid confidence intervals, and (b) an IRT-based adaptive test that infers a latent ability (ε) with far fewer samples by targeting high-information items. Together, these ideas provide a principled, statistically grounded procedure for reporting both point estimates and uncertainty in benchmarked capabilities.

**Strengths:**

**Originality.**

Reframes prompt sensitivity as a systematic bias in capability inference, proving identifiability failures when phrasing is dependently sampled, and proposes random natural perturbations to recover unbiased estimates—an elegant synthesis of CTT/IRT with practical LLM evals.  Introduces two concrete inference methods clustered bootstrapping for accuracy and an IRT-based adaptive test (LAAT) for latent ability—moving beyond prior sensitivity metrics to a principled estimation toolkit.

**Quality.**

Theoretical backbone is clear: defines capability under CTT/IRT, formalizes the bias mechanism, and states propositions establishing non-identifiability and asymptotic guarantees. Methods are operationalized with algorithms and practical guidance (e.g., bootstrap at the question level; LAAT selection via Fisher information), plus discussion of budget/complexity (Neyman allocation).   Empirics are meaningful: multi-benchmark, multi-model studies show large, direction-agnostic bias (up to ±15 p.p.) and sizable within-item variability (MAD typically 10–50 p.p.); SOTA models remain sensitive.

**Clarity.**

Exposition cleanly separates theory, inference and evidence: introduces CTT vs. IRT background before deriving models with perturbation terms and then presenting algorithms/figures that trace bias and sensitivity. Definitions are crisp and practical (e.g., the construction of natural perturbations and the MAD metric), which makes the abstract claims directly testable and reportable.

**Significance.**

Addresses a central pain point in LLM evals, unstable scores and misleading leaderboards, by restoring statistical inference to benchmarking and giving intervals and sensitivity as first-class outputs.Broad impact: provides a template labs can adopt now (perturbation-aware CIs; MAD reporting) and a path to more sample, efficient, capability-faithful testing via LAAT, especially valuable as models scale yet remain brittle on frontier tasks.

**Weaknesses:**

1. “Natural perturbations” are underspecified and may import generator bias. The paper lists categories (instruction rewording, option reordering, prompt paraphrase) and suggests using a strong LLM or rules to produce perturbations, but gives no standardized taxonomy, coverage metric, or validity checks; recovery of ε relies on perturbations being “almost” i.i.d. draws from the phrasing space, which is hard to verify. Different generators (or seeds) could tilt both bias estimates and MAD, confounding cross-paper comparisons.

2. Open-source experiments use two benchmarks (LMEntry, BBH) and 8 subtasks; SOTA results cover only MR and GPQA—and only MR gets full treatment due to budget; GPQA perturbations are produced by gpt-4.1-mini, potentially entangling generator and evaluatee. Claims about leaderboard distortion and sensitivity may not generalize to code, tool-use, or long-context tasks; using one vendor model to generate test variants for another poses subtle dependence risks.

3. Building a benchmark is a time-consuming and expensive process. For example, CBA require to expand the question of benchmark to different formal by a LLM, which is time-consuming and expensive. And the result of the LLM still need to be checked by human, which is also time-consuming and expensive.

4. LAAT assumes a specific IRT form with item difficulty/discrimination known a priori; ability is updated via Fisher information and Newton updates. If the true process deviates from the logistic 2PL with an additive phrasing term, estimates are misspecified; moreover, item parameters must be “properly calibrated.” Ability rankings and standard errors may be brittle to the prior, to miscalibrated items, or to departures from the assumed link.

**Questions:**

1. How to deal with the extreme situation? For example, LLM is able to answer the complex arithmetic problems, but in past long time, fail to answer "which is bigger, 9.11 and 9.9".

2. For section 5.3, Could authors provide result on larger model or closed-source model?

3. Could author provide more vistualization samples of question in Main text？

---

> ### Author Response · Authors · 2025-11-20
>
> Thank you for the helpful comments. We appreciate the reviewer’s engagement and believe that several of the concerns stem from a misunderstanding of what the paper aims to contribute. Our goal is not to introduce novel algorithms but to formalize a framework for AI evaluation as statistical inference grounded in a theory of capability. The methods we present are intentionally simple, because they serve only as proofs-of-concept to illustrate how the framework operates.
>
> In the revision, we have **restructured the text to foreground the conceptual contribution** and clarify how existing tools naturally fall out once a capability model is made explicit. Below we address the specific concerns.
>
> **Weaknesses**
>
> P1. “Natural perturbations” is indeed left vague, and this is a fundamental limitation of any evaluation framework involving natural language variation. **Our contribution is to make this assumption explicit rather than implicit.** In our framework, perturbations define a phrasing distribution, and inference is only as reliable as the evaluator’s choice of that distribution. Section 6.1.2 now clarifies that any perturbation method induces its own phrasing space and that verifying its alignment with a “true” distribution is impossible. This is precisely why evaluations should be framed as inference, with assumptions stated transparently.
>
> P2. We have calibrated the scope of our empirical claims. The SOTA experiments are moved to the Appendix, where they are framed as probes that illustrate inconsistent inferences when perturbation sensitivity is ignored. Additional results on other benchmarks (Appendix F.2) are included, though we emphasize that these experiments are demonstrations of the conceptual issue, not a comprehensive empirical study.
>
> P3 & 4. Yes, benchmark creation is indeed expensive, and our methods require even more time/effort. **This is precisely the central theme of our framework**: more robust evaluation necessarily requires either (i) more data (e.g., calibrated item parameters or perturbations) or (ii) stronger modeling assumptions. There is no free lunch. Table 3 and the discussion (lines 458-464) now emphasize these trade-offs and situate LAAT (IRT) as one possible instantiation among many.
>
> In the discussion section (lines 475-485), we also point out that any theory of capability is a choice, and that there is no “correct” theory; that decision depends on the evaluator as long as they make them explicit in the benchmark. Future work should investigate AI’s cognitive behaviors to better inform the choice of capability theory in evaluations.
>
> **Questions**
>
> Q1. This gets at a related problem that LLMs make idiosyncratic mistakes that do not necessarily scale to question difficulty. In our model, one could explain that away by adding another dimension to any of the latent ability models, though we suspect that these mistakes in practice are noise and would still fall well under a one-dimensional ability model.
>
> Q2 and 3. Note again that we have relegated these empirical results in the Appendix to focus on our conceptual argument. We can provide these details in the Appendix, though these experiments are very much tangential to our central contributions.

---

> > ### Comment · Reviewer_rz2e · 2025-11-27
> > **Replying to Rebuttal**
> >
> > Thank for the response. I have no further concerns, The paper's reflections on benchmarks are meaningful. But considering its limited practicality for constructing a benchmark, I keep my socre.

---

> > > ### Author Response · Authors · 2025-11-27
> > >
> > > Thank you for the response to our review! What precisely is the limited practicality that the reviewer is referring to?
> > >
> > > If it’s the additional data assumptions required in the IRT (and CDM, BSNM models) — the methods we introduce are quite standard and make mild assumptions. For example, there are numerous IRT-based models (all of which fall under our framework) that have been published in similar venues (e.g., Zhuang et al., 2023; Maia Polo et al., 2024), despite the additional legwork required. **More complicated methods do not necessarily mean they are not practical!**
> > >
> > > If it’s the usefulness of the conceptual framework — since the reviewer agrees that the point we make about benchmarks is meaningful, what about our framework is not practical? **The goal of our paper is to help future researchers ground their evaluations in more solid theory and inference procedures.**

---

### Official Review · Reviewer_fvvW · 2025-11-03

**Soundness:** 3
**Presentation:** 4
**Contribution:** 2
**Rating:** 6
**Confidence:** 4

**Summary:**

This paper proposes a principled framework for evaluating AI capabilities as inference, reframing benchmark evaluation not as simple measurement but as statistical inference grounded in a theory of capability. Drawing inspiration from psychometrics, the authors bridge Classical Test Theory (CTT) and Item Response Theory (IRT) with modern AI benchmark evaluation. They demonstrate that current benchmarks violate independence assumptions and induce systematic bias due to prompt sensitivity.

**Strengths:**

1.	Proposes two well-defined, generalizable methods (CBA, LAAT) with theoretical proofs and empirical validation.
2.	Quantifies systematic bias from benchmark phrasing; introduces sensitivity metric (MAD) for reporting uncertainty.
3.	Comprehensive experiments across 7 open-source and 2 proprietary models on multiple benchmarks (BBH, LMentry, GPQA).

**Weaknesses:**

1.	The framework currently addresses only one confounding factor (prompt sensitivity). Other realistic confounders—hyperparameters, context windows, data contamination—are acknowledged but not empirically integrated.
2.	While philosophically appealing, the notion of “AI capability as latent ability” might require stronger empirical validation or correlation with real-world task performance.
3.	Some proofs (e.g., identifiability under dependent phrasing) are deferred to the appendix; including brief intuitions in the main text would improve accessibility.

**Questions:**

1.	How sensitive are the conclusions to the choice of perturbation generation model？
2.	Could the proposed framework generalize beyond text-based LLMs (e.g., vision-language or reinforcement learning benchmarks)?

---

> ### Author Response · Authors · 2025-11-20
>
> Thank you for the kind comments!
>
> **Weaknesses**
>
> P1. Indeed, the current instantiation of our framework focuses on a single confounding factor, prompt sensitivity. Our aim in this paper is to provide a proof-of-concept for the broader idea that evaluations should be framed as inference from a capability model, rather than direct measurement. Prompt sensitivity offers a clean setting in which we can formalize violations of standard assumptions and derive corrections. We now clarify in the revision that our framework is designed to be modular: additional confounders such as hyperparameter variance, context-length effects, or contamination can be incorporated by extending the latent ability model. We are actively working on such extensions and hope that this paper lays the conceptual foundation for a more comprehensive theory.
>
> P2. We appreciate the suggestion for stronger empirical grounding for the claim that “AI capability” acts as a latent ability. In the revision we emphasize that our goal is not to equate AI capability with human ability, but rather to borrow the statistical logic of latent variable models to formalize uncertainties that arise when mapping observed performance to underlying skills. Our experiments demonstrate that failing to account for even a single type of perturbation leads to systematically biased estimates. This illustrates why a latent-variable perspective is necessary for robustness and reliability, even if the full construct validity of “AI capability” will require continued investigation.
>
> P3. Thank you for pointing this out. All theoretical results in the main text are now accompanied by intuitive explanations summarizing why the propositions hold, how dependence affects identifiability, and what parts of the inference pipeline are most sensitive to perturbation structure. We hope this improves accessibility without sacrificing rigor.
>
> **Questions**
>
> Q1. The inferences will certainly shift, and Proposition 1 in the new draft partially addresses this question. In short, they depend on the distance between the true distribution $\mathcal P$ and the different perturbed distributions.
>
> Q2. Yes—the framework is not specific to text-based LLMs. We intentionally use the term “AI capability” to emphasize generality. The key components of our approach are (i) a latent ability model and (ii) a structured treatment of perturbations that break benchmark assumptions. Both exist in RL (e.g., stochastic dynamics, initialization variance), vision-language tasks (e.g., multimodal fusion errors, prompt-template sensitivity), or even systems-oriented evaluations. We have added text (lines 113-199 and 187-190) outlining how the same inferential perspective applies to robustness issues in RL and multimodal evaluation.

---

### Note · Program_Chairs · 2026-01-17
**Submission Desk Rejected by Program Chairs**

The following references in this submission do not refer to real documents and/or have major errors in bibliographic information:

 Yue Dong et al. A psychometric approach to evaluating deep learning models. Knowledge-Based Systems, 2021.
Mostafa Dehghani et al. Benchmarking neural network robustness to distribution shifts. In $I C L R$, 2021.
Yue Dong et al. A neural irt model for automatic question difficulty estimation. In IJCAI, 2020.
H. Burdick et al. A psychometric ai evaluation framework based on item response theory. Artificial Intelligence, 2019.
Z. Chen et al. Bayesian psychometric modeling for llm evaluation. arXiv preprint, 2025.
R. Hernandez et al. Psychometric modeling for automated item generation and evaluation. KnowledgeBased Systems, 2021.
X. Wang et al. Hierarchical item response models for large language model assessment. In OpenReview, 2025.